# Worldwide Prevalence of Alcohol Use in Non-Fatally Injured Motor Vehicle Drivers: A Systematic Review and Meta-Analysis

**DOI:** 10.3390/healthcare11050758

**Published:** 2023-03-05

**Authors:** Laura Kassym, Assiya Kussainova, Yuliya Semenova, Almas Kussainov, Damir Marapov, Marat Zhanaspayev, Zhanar Urazalina, Almira Akhmetova, Madi Tokanov, Yerbol Smail, Geir Bjørklund

**Affiliations:** 1Department of General Medical Practice with a Course of Evidence-Based Medicine, NJSC “Astana Medical University”, Astana 010000, Kazakhstan; 2Department of Infectious Diseases, Dermatovenereology and Immunology, NJSC “Semey Medical University”, Semey 071400, Kazakhstan; 3School of Medicine, Nazarbayev University, Astana 010000, Kazakhstan; 4Department of Psychiatry and Narcology, NJSC “Astana Medical University”, Astana 010000, Kazakhstan; 5Department of Public Health, Economics and Health Care Management, Kazan State Medical Academy—Affiliate of Russian Medical Academy of Professional Education of the Ministry of Health of Russia, Kazan 420000, Russia; 6Department of Traumatology and Pediatric Surgery, NJSC “Semey Medical University”, Semey 071400, Kazakhstan; 7Department of Emergency Medicine, NJSC “Semey Medical University”, Semey 071400, Kazakhstan; 8Republican Scientific and Practical Center for Mental Health, Pavlodar 140017, Kazakhstan; 9Council for Nutritional and Environmental Medicine, 8610 Mo i Rana, Norway

**Keywords:** alcohol, prevalence, injured, drivers, meta-analysis

## Abstract

Drunk driving is an important risk factor significantly contributing to traffic accidents and their associated lethality. This meta-analysis of observational studies aims to provide the estimates of drunk driving prevalence in non-lethally injured motor vehicle drivers in relation to the world region, blood alcohol concentration (BAC), and quality of the primary study. A systematic search for observational studies that examined the prevalence of drunk driving in injured drivers was performed, and 17 studies comprising 232,198 drivers were included in the pooled analysis. The pooled prevalence of drunk driving in injured drivers was found to be 16.6% (95% CI: 12.8–20.3%; I^2^ = 99.87%, *p* < 0.001). In addition, the prevalence of alcohol use ranged from 5.5% (95% CI: 0.8–10.1%) in the Middle East, North Africa, and Greater Arabia region to 30.6% (95% CI: 24.6–36.5%) in the Asia region. As for the subgroups with different thresholds of BAC, the maximum value of 34.4% (95% CI: 28.5–40.3%) was found for a dose of 0.3 g/L. The prevalence of alcohol use reported by high-quality studies was 15.7% (95% CI: 11.1–20.3%), compared to 17.7% (95% CI: 11.3–24.2%) reported by studies of moderate quality. These findings could inform law enforcement efforts to promote road safety.

## 1. Introduction

Road traffic injuries constitute a major global health problem, being a leading cause of death for individuals aged 5–29 years. As many as 1.35 million people die in car crashes every year and another 20–50 million people suffer non-lethal injuries, many of which resulting in disabilities. Pedestrians and two-wheelers (cyclists/motorcyclists) are the most vulnerable to traffic accidents as over 50% of deaths are attributed to these categories of road users. Although not more than 60% of all vehicles belong to developing countries, they account for 90% of all global road lethality. There is a large between-country variability, but the African region has the highest rate of traffic fatality, while the European region has the lowest [1].

Drunk driving is an important risk factor significantly contributing to traffic accidents and their associated lethality. This is a major global health problem that has serious public effects. It refers to the act of driving while under the influence of alcohol, which impairs a driver’s ability to operate a vehicle safely [1]. Alcohol consumption leads to mental confusion, impaired binocular vision, slowed reaction time, and reduced attention, in a dose-dependent manner, which seriously affects driving skills [2]. Thus, alcohol consumption increases the risk of traffic accidents in a dose-dependent manner. This risk elevates already low blood alcohol concentrations (BAC) and reaches significant levels when BAC exceeds 0.05 g/dL, which is equivalent to two standard drinks in the first hour [1]. Moreover, alcohol consumption may lead to a subjective perception of safe driving capacity in individuals with a history of driving under the influence, which can prompt them to drive while intoxicated [3]. As a result, from 5 to 35% of all road deaths globally have been estimated to result from drunk driving [4]. 

In response to this problem, many countries have strengthened law enforcement efforts to promote road safety. According to the World Health Organization (WHO), it is a best practice to set BAC limits of 0.02 g/dL for commercial drivers who transport people, and also for young and novice drivers. As for the general population, the WHO recommends setting BAC limits of 0.05 g/dL. Overall, 45 countries follow the WHO recommendations and the majority belong to high-income strata. However, law enforcement efforts are inadequate in many developing countries or are perceived as non-obligatory by local drivers [1]. 

Although essential for understanding the problem’s magnitude, data on drunk driving are limited and there is no current evidence emerging from a pooled analysis. Therefore, this systematic review and meta-analysis of observational studies is aimed at providing the estimates of drunk driving prevalence in non-lethally injured motor vehicle drivers in relation to the world region, BAC, and quality of study.

## 2. Materials and Methods

### 2.1. Protocol and Registration

A review protocol was not registered in any database before the initiation of this study.

### 2.2. Eligibility Criteria

The following inclusion criteria were applied: (i) the study reported prevalence of alcohol impairment; and (ii) the study included injured motor vehicle drivers. No filters for year of publication, age, or gender were applied. We excluded studies with the following characteristics: (i) publications that included fatally injured motor vehicle drivers; (ii) articles that described only other categories of road users (cyclists, pedestrians, passengers, etc.); (iii) studies that included data on specific anatomical sites of trauma only (maxillofacial, head, thoracic, pelvic, spinal column, etc.); (iv) studies that did not report prevalence rates or did not contain sufficient data for calculation of prevalence rate; (v) studies of low quality based on the findings of quality assessment; and (vi) published articles in a language other than English.

### 2.3. Search Strategies

Searches of three academic databases (PubMed, Google Scholar, and Research Gate) were systematically carried out up to 30 November 2022. A combination of the keywords “alcohol”, “drunk driving”, “injury”, “trauma”, “road”, and “traffic” was used. A strategy for PubMed included (alcohol OR alcohol drinking OR drunk driving) AND (injury OR trauma) AND (road OR traffic). After that, we screened the abstracts of all identified publications to determine if they met the inclusion criteria. Finally, we checked the reference lists of all eligible articles to find additional relevant articles.

### 2.4. Study Selection

The initial search and selection of articles was performed independently by two re-viewers (A.K and L.K.), who screened for titles and abstracts and excluded all articles that did not meet the inclusion criteria. Subsequently, we retrieved the full texts of articles that were considered to be eligible and evaluated all studies on the basis of their design. Any differences of opinion on study eligibility were resolved in discussions with Y.S. and G.B. The selection process following PRISMA guidelines is presented in Figure 1.

### 2.5. Data Extraction

After selecting the articles, the required information was extracted and entered in a standardized form: first author, publication year, location of the study, study design, sample size, prevalence, age, test used, and blood alcohol concentration level (g/L). In some studies, we calculated the sample size of injured motor vehicle drivers and prevalence of alcohol use among them using raw data from the tables, figures, and text of the articles [5,6,7,8,9,10,11,12,13,14,15].

### 2.6. Quality Assessment

The quality of the studies was assessed according to the Joanna Briggs Institute (JBI) checklist for prevalence studies [16]. This tool contains nine questions with four responses, including “yes”, “no”, “unclear”, and “not available”. Appropriate sample frame, study participants sampled, adequacy of sample size, description of study subjects and setting, sample size justification, sufficiency of data analysis, validity of used methods, reliability of measurement methods, appropriateness of statistical analysis, and adequacy of response rate were utilized to assess the risk of bias. The number of positive responses proceeded, and subsequently, the studies were subdivided into three groups: low quality (scores 1 and 2 out of 9), moderate quality (scores 3–6 out of 9), and high quality (scores 7–9).

### 2.7. Data Analysis

We carried out statistical analysis using the OpenMeta[Analyst] (Brown University, Providence, Rhode Island, United States of America) and JASP version 0.16.4 (University of Amsterdam, Amsterdam, The Netherlands). The random effects model was used and the restricted maximum likelihood method was applied to calculate the prevalence of alcohol use among non-lethally injured drivers with 95% confidence intervals (CI). The heterogeneity index (I^2^) value was computed to assess heterogeneity among studies and *p*-value was determined using the chi-square test to assess statistical significance of heterogeneity.

Subgroup analyses were provided when the prevalence was stratified by region, the level of BAC, and quality of the primary study.

## 3. Results

The process of selecting papers using the PRISMA tool is demonstrated in Figure 1. We found 1213 relevant articles via the PubMed, Google Scholar, and ResearchGate databases. Due to duplicate records and irrelevant content, 1129 papers were removed. A total of 17 articles out of 84 were included in the final meta-analysis after the eligibility assessment and evaluation of the methodological quality.

In the current study, we pooled and analyzed data from 17 articles comprising 232,198 injured drivers. We included four studies from North America, three studies from Europe, three studies from South America, two studies from Asia, two studies from the Middle East, North Africa, and Greater Arabia, two studies from Sub-Saharah Africa, and one study from Australia and Oceania. The sample size ranged from 96 to 72,419 injured drivers. The age of the participants ranged from 14 years to 93 years. Almost all studies described the results of the definition of BAC using blood samples. Using the JBI checklist for prevalence studies to evaluate the quality of studies, all the included studies were scored as of moderate or high quality. Table 1 demonstrates the main characteristics of the studies included in the meta-analysis.

### 3.1. Pooled Prevalence

Analyzing the worldwide prevalence of alcohol use among injured drivers using the random effects model, we found the pooled prevalence based on the 17 studies to be 16.6% (95% CI: 12.8–20.3%; I^2^ = 99.87%, *p* < 0.001) (Figure 2).

### 3.2. Subgroup Analyses

When the prevalence was stratified by the world region, we found that the prevalence of alcohol use among injured drivers ranged from 5.5% (95% CI: 0.8–10.1%) in the Middle East, North Africa, and Greater Arabia region to 30.6% (95% CI: 24.6–36.5%) in the Asia region. The statistical heterogeneity of the indicator was found to be high in North America (I^2^ = 99.84%, *p* < 0.001), the Middle East, North Africa, and Greater Arabia (I^2^ = 96.88%, *p* < 0.001) and notable in South America (I^2^ = 83.53%, *p* = 0.004) and Asia (I^2^ = 76.21%, *p* = 0.04). Heterogeneity was insignificantly moderate in the Sub-Saharan Africa subgroup (I^2^ = 50.22%, *p* = 0.156) and insignificantly low in the Europe subgroup (I^2^ = 0.17%, *p* = 0.231) (Figure 3).

When we compared the prevalence of alcohol consumption in the subgroups with different thresholds of BAC, the maximum value of 34.4% (95% CI: 28.5–40.3%) was found for a dose of 0.3 g/L. The minimum level of injured drunk drivers of 10.5% (95% CI: 4.2–16.7%) was indicated in the studies where the threshold of BAC was not specified by the authors. However, the dependence of the prevalence of alcohol use among the injured drivers on the threshold concentration was not statistically significant (*p* = 0.245) (Figure 4).

When the prevalence was grouped by quality of study, the prevalence of alcohol use in the participants of high-quality studies was 15.7% (95% CI: 11.1–20.3%), compared to 17.7% (95% CI: 11.3–24.2%) in the injured drivers from studies of moderate quality, but the differences were not statistically significant (*p* = 0.619) (Figure 5).

## 4. Discussion

The aim of our study was to estimate the global prevalence of alcohol consumption in non-lethally injured motor vehicle drivers, which is of ultimate importance for the strengthening of enforcement efforts to promote traffic safety. All the included studies were conducted between 2012 and 2021. Due to the high heterogeneity of the results, a random effects model was utilized in all the pooled analyses. Our meta-analysis of 17 studies demonstrates that the prevalence of alcohol use among injured drivers is 16.6% (95% CI: 12.8–20.3%; I^2^ = 99.87%, *p* < 0.001). Actually, there was substantial heterogeneity between the studies, and subgroups were analyzed to identify the sources of such heterogeneity. The present findings were similar to the results of Borges et al. (2006), who found that the prevalence of drinking within 6 h prior to the injury among 11,536 non-fatally injured cases was 20.9% (*n* = 2406). The lowest prevalence of acute alcohol use varied from 6.3% (WHO–Canada) to 46.4% (WHO–South Africa) [23]. In fact, there is a wide spectrum of primary studies investigating the prevalence of alcohol use among specific categories of road users. However, we failed to find the relevant systematic reviews or meta-analyses committed to study the prevalence of alcohol consumption in fatally injured car or truck drivers. There are few secondary studies dedicated to the abovementioned research question. For instance, a study by Asgarian et al. (2019) showed that the prevalence of alcohol use among fatally injured motorcyclists was 30% [24]. Another study demonstrated that the prevalence of alcohol consumption in fatally injured motor vehicle drivers from the USA was 40.2% [25]. It might be supposed only that the alcohol consumption contributes greatly to the increasing proportion of fatal injuries. 

We found that the difference in the prevalence of alcohol consumption between injured drivers from different world regions was statistically significant (*p* < 0.001). Obviously, the described discrepancies might be explained with the number of factors such as religious, social, and cultural constraints, development level of the healthcare system, and enforcement of law regulating alcohol consumption by drivers. Our study highlighted the lowest level of alcohol consumption of 5.5% (95% CI: 0.8–10.1%) in the injured drivers from the Middle East, North Africa, and Greater Arabia regions, which are recognized as the center of the Islamic world. Alcohol abuse is a well-known public health problem in Western countries, but there is no sufficient data on the magnitude of this problem in Muslim Arab nationals. The prevalence of alcohol use disorders has been shown to not exceed 11.2% in psychiatric patients and 15.5% in a non-clinical sample [26]. The highest prevalence of alcohol use of 30.6% (95% CI: 24.6–36.5%) among injured drivers was detected in the Asia region, including China, Vietnam, and India. Millwood et al. (2013) showed that 33% of men and 2% of women from 10 study areas in China reported drinking at least weekly [27]. A recent meta-analysis of 2870 studies identified that a maximum rate of alcohol drinkers among the Indian population reached 76.1% (95% CI = 68.1–82.6%; males) and 63.7% (95% CI = 49.4–75.7%; females) [28]. We can thus suggest that the prevalence of alcohol use among injured drivers correlates with the number of alcohol drinkers in a general population. 

Our meta-analysis did not identify statistically significant differences in prevalence of alcohol use among injured drivers depending on the thresholds of BAC. The current literature represents the large spectrum of studies devoted to analyzing the impact of BAC level on road safety. For example, the reduction of the legal BAC limit from 0.8 to 0.5 mg per 100 mL of blood did not change the rate of traffic accident or fatality rates [29]. Another study revealed that lowering the BAC from 0.10 to 0.08 g/dL in the United States from 1982 to 2014 showed a 10.4% reduction in annual drinking driver fatal crash rates [30]. One plausible explanation for this variety is the contribution of other multiple factors to fatal or nonfatal crash rates. 

When we compared the prevalence of alcohol consumption in the studies of moderate vs. high methodological quality, no statistically significant differences were found. Despite the high quality of publication as defined by the JBI checklist for prevalence studies, the research article by Carfora et al. (2018) was not included in our meta-analysis. The prevalence of alcohol use among 609 injured drivers was 91.5% (*n* = 557). This extremely high, out-of-range parameter might be explained with the performance of toxicological analyses exclusively in suspicious cases [31]. Another explanation of high prevalence of alcohol use among Italian injured drivers is that Italy was recognized as the country with an overall best practice for drunk-driving laws according to the WHO Global Status Report on Road Safety 2018 [1]. 

There are a number of shortcomings of this systematic review that have to be dis-cussed. The major limitation of our study is the very marked heterogeneity of the included studies. Indeed, the wide range of study designs, the different settings of participants’ enrollment, and the various levels of BAC used were among possible contributors to the large heterogeneity of the present meta-analysis. However, we used the validated tool for the assessment of publication quality to exclude the inappropriate studies. Secondly, there is a shortage of original articles with the required data for the estimation of global alcohol use among non-fatally injured drivers. To overcome this, we extracted the essential data with a subsequent calculation of alcohol consumption prevalences in injured motor vehicle drivers. Thirdly, a limited number of variables was available to perform the sub-group analysis. Due to the lack of data on the distribution of drivers by age groups, gender, ethnicity, study design, severity, and site of injury, the corresponding sub-group analyses were not conducted. Lastly, the variety of the study populations, the different study designs, and the peculiar socio-cultural context may serve as confounders that could affect the high heterogeneity of the results of our study. 

Nevertheless, to the best of our knowledge, this is the first systematic review and meta-analysis studying the global prevalence of alcohol consumption in motor vehicle drivers that was performed within the last decade. The strengths of our study include extensive coverage of the existing literature, the utilization of a valid and trustworthy appraisal tool for study quality, and concern about the potential subgroup effects.

### Policy Implications

Drunk driving is a serious issue that poses a significant threat to public safety. Unfortunately, despite the numerous campaigns aimed at reducing drunk driving, it remains a prevalent problem with devastating consequences for individuals and society as a whole [4]. Research studies on drunk driving have brought to light several implications that are important to consider. These studies have not only helped in raising awareness about the issue but have also helped in developing effective strategies to prevent drunk driving.

It has to be noted that drunk driving is a major issue that affects not only the safety of individuals on the road, but also has social and economic implications. Policymakers have an important role to play in addressing this issue, as they can implement laws and regulations to discourage drunk driving and mitigate its harmful effects. One of the most effective policy interventions in addressing drunk driving is the implementation of strict laws and penalties. This includes setting a legal limit for BAC and enforcing it through sobriety checkpoints and random breathalyzer tests. Additionally, harsher punishments can be imposed, including license suspension, fines, and even imprisonment, for those caught driving under the influence [32]. Such penalties serve as a deterrent to would-be drunk drivers and help to reduce the incidence of drunk driving.

Another important policy intervention is education and awareness campaigns. Policymakers can allocate resources to public service announcements, social media campaigns, and other forms of outreach to educate the public about the dangers of drunk driving. This can include information about the risks of impaired driving, strategies for preventing drunk driving, and resources for getting help with addiction or substance abuse. Approaches that promote responsible drinking, such as designating a driver or using public transportation, can also be effective in reducing the number of drunk driving incidents. By raising awareness about the issue, societal attitudes towards drunk driving can be shifted and safer driving habits can be promoted [33].

In addition to these interventions, there is a need for addressing the underlying causes of drunk driving. This includes improving public transportation systems, expanding access to ride-sharing services, and promoting alternative forms of transportation, such as biking or walking. By making it easier for people to move around without driving, the demand for driving can be reduced and this will indirectly result in safer travel behavior [33].

Moreover, the research and development of new technologies to prevent drunk driving have to be supported. For instance, all vehicles could be required to have ignition interlock devices, which prevent the vehicle from starting if the driver has a BAC above the legal limit. Additionally, smartphone apps that provide information about alternative transportation options or allow users to call for a ride can help prevent drunk driving. In general, the development of new technologies should be encouraged, including autonomous vehicles that may reduce the incidence of drunk driving in the future [34]. 

The algorithm-based feature selection method plays a crucial role in predicting drunk driving incidents. These features are derived from sophisticated mathematical models that analyze large sets of data to identify patterns and make predictions [35]. The algorithm-based feature selection method commonly includes driving behavior data, such as the speed of the vehicle, the number of lane changes, and the time of day the driving is occurring. In addition to these features, machine learning algorithms can also incorporate data from other sources, such as social media activity [36], to identify drivers who may be at risk of driving under the influence of alcohol. By analyzing patterns in a driver’s behavior and identifying risk factors, these algorithms can predict the likelihood of a drunk driving incident and alert law enforcement or other authorities to take action.

Furthermore, the social and economic factors that contribute to drunk driving have to be addressed. This includes addressing issues such as poverty, lack of access to healthcare, and addiction. By investing in social programs that address these issues, policymakers can reduce the prevalence of drunk driving and help to create safer and healthier communities [33]. Finally, it is essential to continue to support research in this area to develop new and innovative strategies to prevent drunk driving and ensure public safety on the road.

## 5. Conclusions

In summary, in this meta-analysis we found that the global prevalence of alcohol use in injured motor vehicle drivers was 16,6% (95% CI: 12.8–20.3%; I^2^ = 99.87%, *p* < 0.001), which is predictably lower than that for fatally injured drivers. Further research should include other variables such as gender, age, ethnicity, and severity of injury for more de-tailed analysis of their impact on alcohol consumption among drivers. Policymakers should enforce measures for the reduction of alcohol use in the population to improve road safety.

## Figures and Tables

**Figure 1 healthcare-11-00758-f001:**
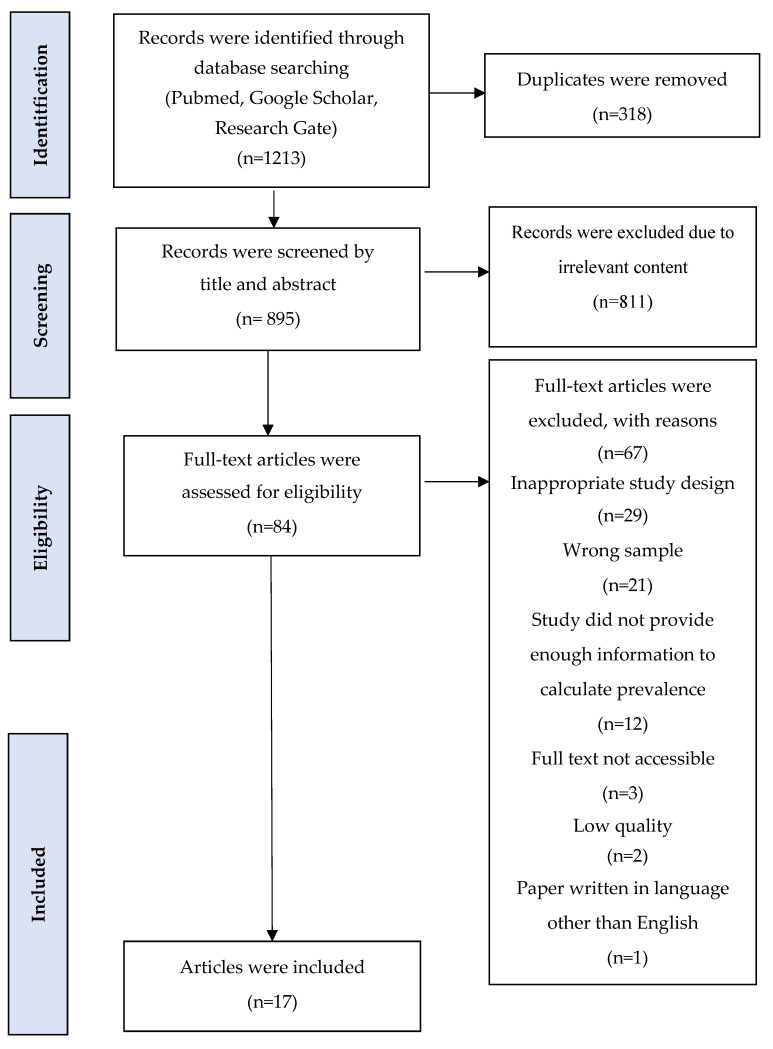
Flow diagram of Preferred Reporting Items for Systematic Reviews and Meta-Analyses (PRISMA) presenting the process of search and selection of studies on prevalence of alcohol impairment in injured motor vehicle drivers.

**Figure 2 healthcare-11-00758-f002:**
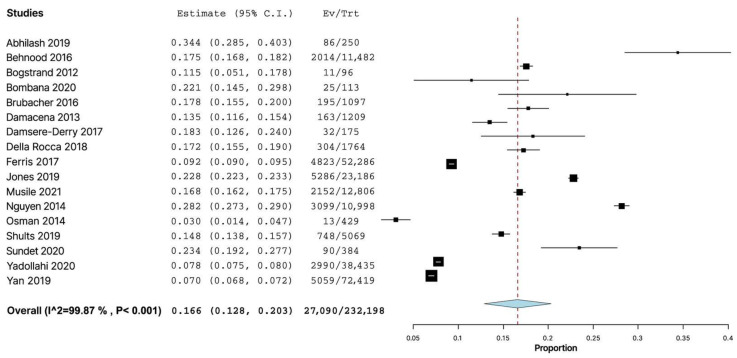
Forest plot of studies (*n* = 17) examining the prevalence of alcohol use among non-lethally injured motor vehicle drivers (*n* = 232,198) [5,6,7,8,9,10,11,12,13,14,15,17,18,19,20,21,22].

**Figure 3 healthcare-11-00758-f003:**
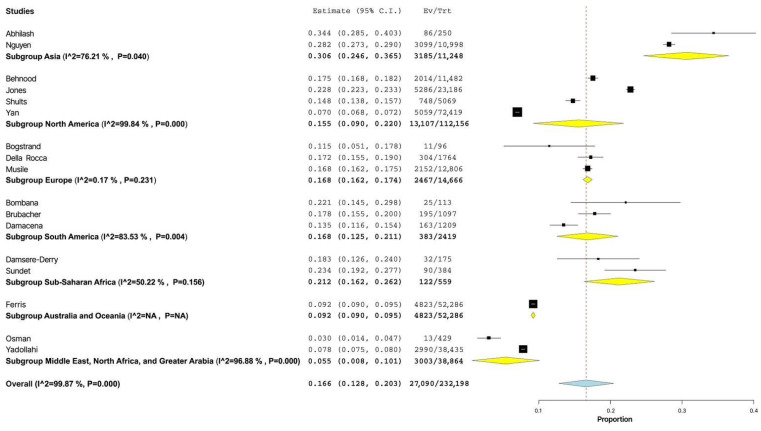
Forest plot of prevalence of alcohol use among non-lethally injured drivers depending on the world region [5,6,7,8,9,10,11,12,13,14,15,17,18,19,20,21,22].

**Figure 4 healthcare-11-00758-f004:**
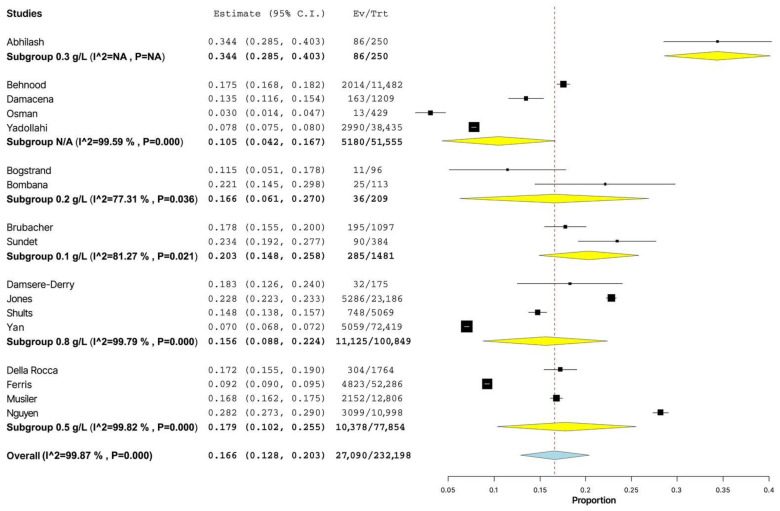
Forest plot of prevalence of alcohol use among non-lethally injured drivers depending on the level of blood alcohol concentration [5,6,7,8,9,10,11,12,13,14,15,17,18,19,20,21,22].

**Figure 5 healthcare-11-00758-f005:**
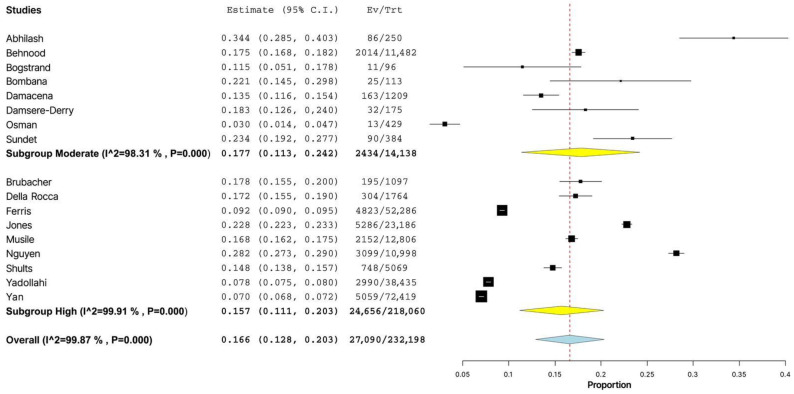
Forest plot of prevalence of alcohol use among injured drivers depending on the study quality [5,6,7,8,9,10,11,12,13,14,15,17,18,19,20,21,22].

**Table 1 healthcare-11-00758-t001:** The main characteristics of 17 studies examining the prevalence of alcohol use in injured patients.

Row	Study	Location	Region	Study Period	Study Design (Defined by Authors)	Sample Size	Prevalence of Alcohol in Injured Persons, % (*n*)	Age Range at Baseline (Years)	Used Test	BAC level (Alcohol Level)	Vehicle Type	Quality
1	Abhilash, 2019 [5]	India	Asia	2018	Retrospective analysis	250	34.4% (*n* = 86)	≥18	Blood sample	>0.3 g/L	Drivers	Moderate
2	Behnood, 2016 [6]	USA	North America	2004–2012	Not reported	11,482	17.54% (*n* = 2014)	NA	NA	NA	NA	Moderate
3	Bogstrand, 2012 [7]	Norway	Europe	2007–2008	Case-control	96	11.5% (*n* = 11)	≥18	Blood sample	>0.2 g/L	Drivers	Moderate
4	Bombana, 2020 [8]	Brasil	South America	2018–2019	Cross-sectional	113	22.12% (*n* = 25)	>18	Blood sample	>0.2 g/L	Car/truck drivers, motor-cyclists	Moderate
5	Brubacher, 2016 [17]	Canada	North America	2010–2012	Prospective cross-sectional	1097	17.8% (*n* = 195)	NA	Blood sample	>0.1 g/L	Car drivers	High
6	Damacena, 2013 [9]	Brazil	South America	2013	Not reported	1209	13.48% (*n* = 163)	>18	NA	NA	Car/van driver, bus driver, truck driver, motorcycle driver	Moderate
7	Damsere-Derry, 2017 [10]	Ghana	Sub-Saharah Africa	2015	Not reported	175	18.3% (*n* = 32)	18–78	Blood sample	>0.8 g/L	Motor-cyclists, drivers	Moderate
8	Della Rocca, 2018 [18]	Italy	Europe	2010–2014	Nor reported	1764	17.23% (*n* = 304)	15–93	Blood sample	>0.5 g/L	Car/van, motor-cycle, truck, bus drivers	High
9	Ferris, 2017 [11]	Australia	Australia and Oceania	2000–2010	Not reported	52,286	9.22% (*n* = 4823)	NA	Blood sample	>0.5 g/L	Driver	High
10	Jones, 2019 [19]	USA	North America	2008–2014	Interrupted times series analysis	23,186	23% (*n* = 5286)	≥16	Urine sample	>0.0 g/L (16–20 YRS) >0.8 g/L (≥21 YRS)	Drivers of cars, trucks of all sizes	High
11	Musile, 2021 [20]	Italy	Europe	2009–2017	Not reported	12,806	16.8% (*n* = 2152)	NA	Blood sample	>0.5 g/L	Drivers	High
12	Nguyen, 2014 [12]	Viet Nam	Asia	2009–2010	Baseline survey	10,998	28.18% (*n* = 3099)	>14	Blood sample	>0.5 g/L	Motor-cycle and car drivers	High
13	Osman, 2014 [13]	United Arab Emirates	Middle East, North Africa, and Greater Arabia	2006–2007	Prospective	429	3.03% (*n* = 13)	15–53	Blood sample	NA (forensic police)	Drivers	Moderate
14	Shults, 2019 [21]	USA	North America	2008–2014	Descriptive report	5069	15% (*n* = 748)	16–20	Blood sample	≥0.8 g/L	Car, truck, van, motorcycle driver	High
15	Sundet, 2020 [14]	Malawi	Sub-Saharah Africa	May 25, 2019 to August 22, 2019	Cross- sectional	384	23.44% (*n* = 90)	≥18	Breathalyzer or a saliva test	>0.1 g/L	Car and pickup drivers, motorcycle drivers	Moderate
16	Yadollahi, 2020 [22]	Iran	Middle East, North Africa, and Greater Arabia	2018	Cross-sectional	38,435	7.78% (*n* = 253)	>15	Blood sample	NA	Car and motorcycle drivers	High
17	Yan, 2019 [15]	USA	North America	2010–2014	Not reported	72,419	6.99% (*n* = 5059)	NA	Blood sample	>0.8 g/L	Motor vehicle driver	High

## Data Availability

Data is contained within the article or supplementary material.

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
