# Peer review of "Worldwide Prevalence of Alcohol Use in Non-Fatally Injured Motor Vehicle Drivers: A Systematic Review and Meta-Analysis"

_healthcare, 2023, doi:10.3390/healthcare11050758_

Round 1

Reviewer 1 Report

This is an interesting topic, the review is well designed and properly presented; although the table and figures require improvement, in particular the figures’ quality is not ideal and it requires to be clearer for the readers. In addition, throughout the text there are spelling mistakes (perhaps word typos generated through a software?) and it requires a thorough search in all the manuscript in order to correct these. Please kindly note the following suggestions for your consideration:

L92. Do you mean strategy?

L94. Correct the word “publications”

Table 1: format the first row of the table, with the description of each column. Also, format the whole table to visually look more neat.

Figure 2: this is a bad quality image, you should rewrite the text in order to be clear and prepare a better quality figure

Figure 3, Figure 4 & Figure 5: same comment as for figure 2  

Check the whole text for these typos, and correct accordingly (see below some examples)

L232: Correct the word “findings”

L240: Correct the word “instance”

L242: Correct the word “prevalence”

L248: Correct the word “obviously”

L249: Correct the word “religious”

L250: Correct the word “enforcement”

L256: Correct the word “psychiatric”

L258: Correct the word “including”

L263: Correct the word “injured”

L266/67: Correct the word “literature”

L272: Correct the word “plausible”

Author Response

Thank you very much for reviewing our manuscript. You have done a great job and we will try to give detailed answers to all comments. The changes made to the manuscript according to your recommendations were marked up using the “Track Changes” function and highlighted in yellow.

Reviewer 2 Report

This systematic analysis is a reliable study conducted following correct procedures. Despite the pitfalls of the high heterogeneity of the results, the shortage of original articles with the required data, and the applicability of only limited variables, the authors identified the global prevalence of alcohol use in injured motor vehicle drivers. In addition, other variables to be added for more detailed analysis in future studies were suggested.

Overall, this is an interesting piece of work deserving the attention of the broad readership of healthcare. However, there are some minor concerns that need to be addressed:

 Minor concerns:

1.    In line 52-53:

For the dose-dependent effects of alcohol intake in traffic accidents, it is recommended that the causal relationship be clearly described, at least as shown in the following example.

e.g., “Alcohol consumption leads to mental confusion, impaired binocular vision, slowed reaction time, and reduced attention, in a dose-dependent manner, which seriously affects driving skills. Thus, alcohol consumption increases the risk of traffic accidents in a dose-dependent manner.”

2.    in line 56-58,

Reference 3 does not fully support the author's claim: "Moreover, alcohol consumption may improve a subjective perception of safe driving capacity prompting people to drive while being intoxicated by alcohol ".

Furthermore, reference 3 cited here does not support the dose effects by statistical differences between the experimental groups.; Young drivers' lower subjective response to intoxication is associated with the decision to drunken driving.

Please clarify the description or cite an appropriate reference.

3.    Image quality needs to be improved in Figures 2, 3, 4, and 5. In particular, the text embedded in the figures is too small and blurry.

Author Response

(The authors gave the same response as above.)

Reviewer 3 Report

The paper presents an interesting survey. Following issues needs to be addressed.

1.       Pls. format your paper according to the journal guideline.

2.       Quantify your results in the abstract

3.       Add a gap analysis section in your introduction and also mention the research questions.

4.       Table 1 presentation can be improved

5.       Pls. add a discussion section listing the implications and interpretation of your survey.

6.       Add open questions to solve in your work   

7.       Following references are missing

a.       Explicit and Size-adaptive PSO-based Feature Selection for Classification

b.       An Effective Genetic Algorithm-Based Feature Selection Method for Intrusion Detection Systems

c.       Orienting Conflicted Graph Edges Using Genetic Algorithms to Discover Pathways in Protein-Protein Interaction Networks

d.       A novel binary chaotic genetic algorithm for feature selection and its utility in affective computing and healthcare

Author Response

(The authors gave the same response as above.)

Round 2

Reviewer 1 Report

Thank you for considering the suggestions and for making the changes. The manuscript has been improved, in addition to the extra section that was added at the end. 

I still believe that Table 1 can be improved, perhaps use the landscape page format in order to have more space for the headings; as you can see the words are broken and not written as a whole. 

Reviewer 2 Report

Thank you for the author's responses. The authors have revised all previous issues raised by the reviewer. Consequently, it made the argument of the article more solid.

Reviewer 3 Report

Revised paper is in a better shape